# Osteocyte-Derived CaMKK2 Regulates Osteoclasts and Bone Mass in a Sex-Dependent Manner through Secreted Calpastatin

**DOI:** 10.3390/ijms24054718

**Published:** 2023-03-01

**Authors:** Justin N. Williams, Mavis Irwin, Yong Li, Anuradha Valiya Kambrath, Brett T. Mattingly, Sheel Patel, Mizuho Kittaka, Rebecca N. Collins, Nicholas A. Clough, Emma H. Doud, Amber L. Mosley, Teresita Bellido, Angela Bruzzaniti, Lilian I. Plotkin, Jonathan C. Trinidad, William R. Thompson, Lynda F. Bonewald, Uma Sankar

**Affiliations:** 1Department of Anatomy, Cell Biology and Physiology, Indiana University School of Medicine, Indianapolis, IN 46202, USA; 2Indiana Center for Musculoskeletal Health, Indiana University School of Medicine, Indianapolis, IN 46202, USA; 3Division of Biomedical Science, Marian University College of Osteopathic Medicine, Indianapolis, IN 46022, USA; 4Department of Biomedical Sciences and Comprehensive Care, Indiana University School of Dentistry, Indianapolis, IN 46202, USA; 5Department of Biochemistry and Molecular Biology, Indiana University School of Medicine, Indianapolis, IN 46202, USA; 6Department of Chemistry, Biological Mass Spectrometry Facility, Indiana University, Bloomington, IN 47405, USA; 7Department of Physical Therapy, School of Health and Human Sciences, Indianapolis, IN 46202, USA

**Keywords:** extracellular calpastatin, Ca^2+^/calmodulin (CaM)-dependent protein kinase kinase 2, osteocytes, osteoclasts, bone remodeling

## Abstract

Calcium/calmodulin (CaM)-dependent protein kinase kinase 2 (CaMKK2) regulates bone remodeling through its effects on osteoblasts and osteoclasts. However, its role in osteocytes, the most abundant bone cell type and the master regulator of bone remodeling, remains unknown. Here we report that the conditional deletion of CaMKK2 from osteocytes using Dentine matrix protein 1 (*Dmp1*)-8kb-*Cre* mice led to enhanced bone mass only in female mice owing to a suppression of osteoclasts. Conditioned media isolated from female CaMKK2-deficient osteocytes inhibited osteoclast formation and function in in vitro assays, indicating a role for osteocyte-secreted factors. Proteomics analysis revealed significantly higher levels of extracellular calpastatin, a specific inhibitor of calcium-dependent cysteine proteases calpains, in female CaMKK2 null osteocyte conditioned media, compared to media from female control osteocytes. Further, exogenously added non-cell permeable recombinant calpastatin domain I elicited a marked, dose-dependent inhibition of female wild-type osteoclasts and depletion of calpastatin from female CaMKK2-deficient osteocyte conditioned media reversed the inhibition of matrix resorption by osteoclasts. Our findings reveal a novel role for extracellular calpastatin in regulating female osteoclast function and unravel a novel CaMKK2-mediated paracrine mechanism of osteoclast regulation by female osteocytes.

## 1. Introduction

Bone is a dynamic tissue, uniquely capable of self-renewing through the process of bone remodeling, which involves the sequential and coupled activities of bone-forming osteoblasts (OBs) and bone-resorbing osteoclasts (OCs) [1]. OBs and OCs in turn are regulated by osteocytes, the most abundant bone cells that originate from matrix-entrapped OBs [2,3]. Osteocytes integrate hormonal and mechanical signals to regulate bone homeostasis. As endocrine cells, osteocytes secrete several key factors including sclerostin (Sost), Dickkopf-1 (DKK1), and osteoprotegerin (Opg), which regulate OB and OC functions [4,5,6]. However, the molecular mechanisms by which osteocytes regulate bone remodeling are not fully understood.

The Ca^2+^/calmodulin (CaM)-dependent protein kinase (CaMK) signaling cascade, initiated by transient increases in intracellular Ca^2+^, involves multifunctional serine/threonine protein kinases CaMKK1 and CaMKK2, and their canonical substrates CaMKI and CaMKIV [7]. CaMKK2 additionally phosphorylates and activates adenosine monophosphate activated protein kinase (AMPK) to coordinate cellular and organismal energy balance [8,9,10,11]. Consequently, inhibition or loss of CaMKK2 in mice protects from diet-induced obesity and insulin resistance [10,11]. CaMKK2 also coordinates inflammatory responses in macrophages and chondrocytes [12,13]. In the skeleton, CaMKK2 plays cell-intrinsic roles in OBs and OCs. Mice lacking CaMKK2 globally (*Camkk2^-/-^*) possess more OBs, fewer OCs and increased bone mass compared to wild-type (WT) mice [14]. Blocking CaMKK2 activity using its selective pharmacological inhibitor STO-609 reverses age-associated decline in trabecular and cortical bone mass and strength, prevents ovariectomy-induced bone loss, and enhances bone fracture healing [15,16,17]. *Camkk2^-/-^* bone marrow-derived progenitors yield higher numbers of alkaline phosphatase-positive OBs in vitro than WT, in part via activation of cyclic adenosine monophosphate (cAMP)-protein kinase A (PKA), and fewer OCs through the downregulation of cyclic adenosine monophosphate (cAMP) response element binding protein (pCREB)—nuclear factor of activated T cells, cytoplasmic (NFATc1) signaling, indicating cell intrinsic roles for CaMKK2 in OBs and OCs [14]. However, the specific roles of CaMKK2 in osteocytes remain unknown.

In this study, we tested the hypothesis that osteocyte-derived CaMKK2 plays an important role in the regulation of bone growth and maintenance using in vivo and in vitro approaches. Our findings reveal a cell-intrinsic role for CaMKK2 in osteocytes in the regulation of OCs in a sex-dependent manner through a novel paracrine mechanism involving secreted calpastatin.

## 2. Results

### 2.1. Conditional Deletion of CaMKK2 in Osteocytes Enhances Trabecular Bone Mass Only in Female Mice

We crossbred *Camkk2^flox/flox^* mice with transgenic mice expressing *Cre* driven by a 8-kb dentin matrix protein 1 (*Dmp1*) promoter to generate heterozygous *Dmp1*-8kb-*Cre*::*Camkk2^flox/WT^* mice which were then mated to generate male and female *Dmp1*-8kb-*Cre*::*Camkk2^WT/WT^* (Control) and *Dmp1*-8kb-*Cre*::*Camkk2^flox/flox^* (*Camkk2^ΔOCY^*) mice [11,18,19] (Figure 1A). *Camkk2* mRNA levels were 13-fold lower in primary *Camkk2^ΔOCY^* osteocytes compared to control osteocytes in both sexes (Figure 1B,C). Deletion of CaMKK2 from osteocytes in female and male *Camkk2^ΔOCY^* mice was also confirmed by immunohistochemistry (IHC; Figure 1D,Di,E,Ei). Conditional deletion of CaMKK2 did not affect osteocyte numbers in either sex (Figure 1F,G).

Examination of distal femora using micro-computed tomography (µCT) revealed a 2.2-fold increase in bone volume fraction (%BV/TV) in 12-week (w)-old female *Camkk2^ΔOCY^* mice compared to age-and sex-matched controls (Figure 1H,Hi). Female *Camkk2^ΔOCY^* long bones also possessed significantly higher trabecular number and lower trabecular separation compared to controls mice, whereas no differences were observed in trabecular thickness (Figure 1Hii–Hiv). In contrast, BV/TV (%) and trabecular bone microarchitecture remained similar between 12-week-old male *Camkk2^ΔOCY^* and control mice (Figure 1I–Iiv). Cortical bone area, cross-sectional thickness, and polar moment of inertia were similar among cohorts of both sex (Appendix A). These data indicate a sex divergent effect on trabecular bone mass by osteocyte-derived CaMKK2.

### 2.2. Ablation of Osteocyte-Derived CaMKK2 Diminishes OCs in Female Mice without Altering OB Function 

We next evaluated whether deletion of CaMKK2 in osteocytes affected bone remodeling. Examination of Von Kossa–McNeil (VKM) stained femur sections revealed significantly higher numbers of OBs in female *Camkk2^ΔOCY^* mice compared to female controls, whereas OB numbers were similar in male *Camkk2^ΔOCY^* and control femurs (Figure 2A,Ai,B,Bi). In contrast, osteoid surface and osteoid thickness remained similar between sex-matched cohorts (Figure 2Aii,Aiii,Bii,Biii). Further, dynamic histomorphometry of double-fluorochrome labeled long bone sections revealed no differences in mineralizing surface, mineral apposition rate or bone formation rate between sex-matched *Camkk2^ΔOCY^* and control mice (Figure 2C–Ciii,D–Diii). Thus, changes in bone formation were not likely the cause of the female-specific increase in cancellous bone mass in female *Camkk2^ΔOCY^*.

Static histomorphometry of tartrate-resistant acid phosphatase (TRAP)-stained bone sections revealed a 1.7-fold reduction in OC numbers and OC surface to bone surface in female *Camkk2^ΔOCY^* mice compared to controls, whereas no such differences were observed in male mice (Figure 2E–Eii,F–Fii). These histomorphometry data revealed a role for osteocytic CaMKK2 in the regulation of OCs in female mice.

### 2.3. Conditioned Media from Female Osteocytes Lacking CaMKK2 Inhibits OC Differentiation and Function 

We surmised that the differential regulation of OCs by female *Camkk2^ΔOCY^* osteocytes occurs through secreted factors. To test this, we isolated primary osteocytes from male and female control and *Camkk2^ΔOCY^* long bones and evaluated the ability of the respective conditioned media (CM) to support OB and OC differentiation by primary wildtype (WT) bone marrow (BM)-derived cells (Figure 3A). We observed no differences in the ability of male or female, control or *Camkk2^ΔOCY^* osteocyte CM to support OB differentiation by WT BM-derived mesenchymal stem cells (MSCs) in vitro as evidenced by alkaline phosphatase or alizarin red staining intensities (Figure 3B,C). In contrast, WT BM cells exposed to CM from female *Camkk2^ΔOCY^* osteocytes yielded 1.7-fold fewer multinuclear TRAP-positive OCs than cells receiving female control CM (Figure 3D,Fi). On the other hand, WT BM cells yielded similar numbers of TRAP-positive multinuclear OCs when treated with male control or *Camkk2^ΔOCY^* osteocyte CM (Figure 3D,Gi). Next, to assess the effects of osteocyte CM on OC function, we plated WT BM cells on hydroxyapatite-coated wells in the presence or absence of osteocyte CM and assessed resorption. BM cells that received female *Camkk2^ΔOCY^* osteocyte CM formed 3-fold fewer resorption pits and resorbed 3-fold lower hydroxyapatite area than cells exposed to control CM (Figure 3E,Fii,Fiii), whereas no such differences were observed when BM cells were treated with male *Camkk2^ΔOCY^* or control CM (Figure 3E,Gii,Giii).

### 2.4. Extracellular Calpastatin Is Enriched in Female CaMKK2-Deficient Osteocyte Conditioned Media Compared to Control and Its Sequestration Relieves OC Inhibition 

It is well documented that osteocytes regulate OC differentiation through the altered expression of receptor activator of nuclear factor κ-B ligand (Rankl) and Opg [20,21]. However, we found no significant differences in *Rankl or Opg* mRNA levels or their ratio in male and female *Camkk2^ΔOCY^* osteocytes compared to controls (Appendix A), indicating the presence of another mechanism. To identify OC-inhibitory factors secreted by CaMKK2-deficient female osteocytes, we concentrated serum-free CM harvested from male and female control and *Camkk2^ΔOCY^* osteocytes and performed proteomic analysis of the secretome using mass spectrometry (LC-MS/MS) (Figure 4A). The experiments were conducted in triplicate and to initially assess the underlying reproducibility and accuracy of the data, we conducted principal component analysis. This revealed grouping of experimental replicates from the same cohort, and divergence of male and female osteocyte CM samples (Figure 4B), consistent with reproducible proteomic data. Although *Camkk2^ΔOCY^* male CM samples displayed more overall variability than other groups within the principal component analysis, they still clustered with each other and apart from the other groups.

Mass spectrometry detected a total of 1182 proteins, including 9293 unique peptides, in male and female osteocyte CM. Of these, 236 proteins were upregulated, 108 proteins were downregulated in male and female *Camkk2^ΔOCY^* osteocyte CM compared to sex-matched controls, and 117 proteins were significantly altered only in female *Camkk2^ΔOCY^* secretome (Figure 4C). Functional annotation analysis confirmed association of the majority of detected proteins with extracellular compartments, including exosomes, vesicles, and extracellular matrix (ECM; Figure 4D), confirming that our CM primarily consisted of secreted and cell surface associated proteins. Reactome pathway analysis identified several pathways associated with degradation of ECM, protein metabolism, and Golgi-associated vesicle biosynthesis and anterograde transport to be significantly altered selectively in female *Camkk2^ΔOCY^* osteocyte CM (Table 1).

Of the proteins associated with ECM degradation, calpastatin, the specific inhibitor of Ca^2+^-dependent cysteine proteases called calpains, was of particular interest as its activity is regulated by Ca^2+^; it possesses CaM-binding domains; and its target calpain has crucial roles in OCs [22,23,24,25]. Though predominantly intracellular, calpastatin and calpains are also secreted molecules [26,27]. Calpastatin was among a group of 62 proteins that were highly enriched in CM from female *Camkk2^ΔOCY^* osteocytes, but not detected in CM from female control osteocytes. On the other hand, levels of calpastatin in male *Camkk2^ΔOCY^* osteocyte CM were not significantly different from those in male control CM (Figure 4E,Ei,F,Fi). Enzyme-linked immunosorbent assay (ELISA) of CM indicated a 3-fold increase in secreted calpastatin in female *Camkk2^ΔOCY^* CM compared to female control CM (Figure 4G). On the other hand, extracellular calpastatin levels in male *Camkk2^ΔOCY^* osteocyte CM were only slightly higher than those in control male CM, and the differences were not statistically significant (Figure 4G). Of note, the female control CM possessed the least amount of extracellular calpastatin of all four cohorts. We next examined intracellular levels of calpastatin in these osteocytes and found that both male and female *Camkk2^ΔOCY^* osteocytes possessed 3.7–3.9-fold higher intracellular calpastatin compared to sex-matched controls (Figure 4H–Hii). Thus, whereas intracellular calpastatin is elevated in CaMKK2-deficient osteocytes of both sexes, the differential increase in extracellular calpastatin was observed only in CM from CaMKK2-deficient female osteocytes.

### 2.5. Osteoclast Inhibitory Effects of Recombinant Non-Cell Permeable Calpastatin Is Sex-Dependent

To understand whether extracellular calpastatin inhibited OC differentiation and function, we treated male and female WT BM cells undergoing RANKL-mediated OC differentiation with varying doses of recombinant human calpastatin domain I, which is non-cell-permeable (NCP) [28,29]. Treatment of female WT BM cells with 0.5, 1.0 and 5.0 µM NCP-calpastatin elicited a 1.7, 1.9 and 3.7-fold decrease in resorption pit number and area resorbed, compared to untreated cells (Appendix A). In contrast, only 5.0 µM NCP-calpastatin caused a significant 2-fold reduction in resorption pit number and area resorbed by male WT BM cells undergoing RANKL-mediated OC differentiation, compared to untreated cells (Appendix A). On the other hand, NCP-calpastatin elicited a dose-dependent reduction in the number of multinuclear OCs produced by male and female WT BM cells (Appendix A). Of note, treatment with 5.0 µM NCP-calpastatin resulted in smaller and fewer OCs and resorption pits in cells from either sex. Taken together, our results indicate that, whereas extracellular NCP-calpastatin inhibited OC differentiation in both sexes, it inhibited OC function, only in females.

### 2.6. Extracellular Calpastatin in Female Camkk2^ΔOCY^ CM Regulates Bone Resorption by Osteoclasts

To further investigate the OC-inhibitory role of secreted calpastatin, we incubated female *Camkk2^ΔOCY^* osteocyte CM with anti-calpastatin antibody or control IgG and assessed the ability of calpastatin-depleted or IgG-incubated CM to support RANKL-mediated OC differentiation by WT BM cells (Figure 5A–C). Consistent with our previous results (Figure 3D,F), we observed 1.8-fold fewer OCs and a 2.8-fold reduction in the number of resorption sites and area resorbed when WT BM cells were treated with IgG-incubated female *Camkk2^ΔOCY^* osteocyte CM (Figure 5C,Ci,D–Dii). Depletion of calpastatin fully reversed the inhibition of OC resorption by female *Camkk2^ΔOCY^* CM (Figure 5D–Dii).

The main function of calpastatin is calpain inhibition. Secreted calpains cleave ECM components to enable cell spreading, ECM attachment and migration in other cell types [28,29]. Since female *Camkk2^ΔOCY^* osteocyte CM inhibits OC differentiation, we surmised that extracellular calpastatin present in the CM blocks migration of osteoclast progenitors towards each other. We performed in vitro monolayer scratch assays to test this and found WT BM cells treated with control CM as well as IgG- or calpastatin-depleted *Camkk2^ΔOCY^* CM to display similar rates of migration (Appendix A), indicating that extracellular calpastatin does not inhibit cell migration.

We next assessed whether calpastatin in the CM affects formation of actin ring-mediated sealing zone that is required for attachment of OCs to bone matrix. WT-BM-derived OCs treated with female control CM, IgG-exposed or calpastatin-depleted *Camkk2^ΔOCY^* CM were stained with rhodamine-phalloidin to detect filamentous (F) actin. OC precursors treated with female control osteocyte CM formed multinuclear OCs with large distinct F-actin rings on glass coverslips (Figure 5E). In contrast, OCs treated with IgG-incubated female *Camkk2^ΔOCY^* CM were smaller with poorly defined F-actin rings, whereas depletion of calpastatin resulted in larger OCs with more distinct F-actin rings (Figure 5E). Taken together, our data indicate a novel paracrine role for osteocyte-derived CaMKK2 in the regulation of OC differentiation and function through a secreted calpastatin-mediated mechanism.

## 3. Discussion

The objective of the current study was to investigate a cell-intrinsic role of CaMKK2 in osteocytes, the master regulators of skeletal homeostasis [6]. Conditional deletion of CaMKK2 from osteocytes elicits a sex-dependent effect on the skeleton. Specifically, we observed enhanced bone mass coupled with fewer OCs in female but not male *Camkk2^Δ^^OCY^* mice. Further, CM isolated from female *Camkk2^Δ^^OCY^* osteocytes suppressed OC formation and function. Calpastatin, a specific inhibitor of Ca^2+^-activated calpains, was highly enriched in female *Camkk2^Δ^^OCY^* CM, its intracellular levels were significantly elevated in male and female *Camkk2^Δ^^OCY^* osteocytes whereas its extracellular levels were differentially altered only in female *Camkk2^Δ^^OCY^* CM. Levels of extracellular calpastatin were significantly lower in female control osteocyte CM compared to males, and the reason for this suppression is not clear. Immunodepletion of calpastatin attenuated the inhibition of OC function by female *Camkk2^Δ^^OCY^* CM. Moreover, treatment of WT BM-derived myeloid cells with exogenous non-cell permeable calpastatin caused a more pronounced inhibitory effect on OC resorption in female-derived cells than male. Based on these cumulative data, we propose that CaMKK2 is an inhibitor of calpastatin expression in osteocytes and that osteocyte-secreted calpastatin blocks OC function in a sex-specific manner (Figure 5F). Thus, our studies identify a novel sex-specific paracrine role for osteocyte-derived extracellular calpastatin in the regulation of OCs.

Calpastatin is the chief inhibitor of Ca^2+^-dependent cysteine proteases called calpains [22]. Calpains proteolyze several intracellular substrates to critically regulate a multitude of cellular processes such as cell motility, spreading, and adhesion to ECM in multiple cell types including OBs and OCs [24,25,30,31,32,33]. In OCs, cleavage by intracellular calpains of their substrates enriched in actin ring, such as talin, filamin A, and Pyk2, is crucial for OC motility, spreading, sealing zone formation, and bone resorption [32,34]. Though predominantly intracellular, extracellular calpains that cleave ECM components have been reported in several systems including OBs, hypertrophic chondrocytes, healing bone fracture calluses, and synovial fluid from rheumatoid arthritis (RA) and osteoarthritis (OA) patients [35,36,37,38,39,40,41]. Calpastatin is also a secreted molecule found in the synovial fluid of RA and OA patients, plasma of patients with pulmonary arterial hypertension, and exosomes from luminal fluid of ovine uterus [28,42,43]. In this study, we demonstrate for the first time that calpastatin is secreted by osteocytes and that extracellular calpastatin acts to regulate the formation and function of female OCs.

Mammalian calpastatin is encoded by a single gene (*Cast*), but the use of multiple promoters and alternative splicing mechanisms leads to the generation of many calpastatin isoforms that vary in molecular mass from 17.5 kDa to 85 kDa [44,45]. Multiple calpastatin isoforms are often present in the same tissue and even cell type. Calpastatin protein consists of an N-terminal L domain devoid of inhibitory activity and four repetitive inhibitory domains (I–IV) that bind to and inhibit calpain in a Ca^2+^-dependent manner [44,45]. We observed two intracellular calpastatin isoforms, a predominant 120 kDa species and a minor 80 kDa species in murine osteocytes through immunoblotting, and both isoforms were enhanced in CaMKK2-deficient osteocytes, regardless of biological sex (Figure 4H). Though the size of the osteocyte-secreted calpastatin is unknown, we surmise that it is the 80 kDa isoform since the lacunar–canalicular transport has a molecular cut off limit of 70–80 kDa [46,47]. Further, phosphorylation of calpastatin by Ser/Thr kinases protein kinase A or protein kinase C decreases its efficiency of calpain inhibition [22]. Being a Ser/Thr kinase, CaMKK2 potentially regulates calpastatin levels and/or activity via phosphorylation, and its absence could enhance the levels of both intracellular and extracellular calpastatin in both sexes.

Extracellular calpastatin is non-cell permeable [28]. Accordingly, exposure of OCs to female osteocyte CM did not alter intracellular levels of calpastatin, calpain substrate talin or calpain activator PKA (Appendix A). How might osteocyte-secreted calpastatin regulate OCs? The only known function of calpastatin is calpain inhibition, and others have postulated that the regulatory effects of secreted calpastatin mainly involve the inhibition of extracellular calpain [28,48]. Secreted calpains cleave ECM components to facilitate the disengagement of α_V_β_3_ integrins with ECM and enable cell migration [28,29]. However, migration of OC precursors is not influenced by extracellular calpastatin (Appendix A). On the other hand, WT BM-derived myeloid progenitors treated with calpastatin-containing female *Camkk2^ΔOCY^* osteocyte CM formed fewer functionally deficient OCs that were also smaller with indistinct F-actin rings, consistent with our in vivo data from female *Camkk2^ΔOCY^* long bones. Further, depletion of calpastatin from the CM completely reversed the inhibition of OC function by female *Camkk2^ΔOCY^* osteocyte CM. Although the mechanism by which extracellular calpastatin regulates OCs is unclear, we hypothesize that osteocyte-secreted calpastatin inhibits OCs either by inhibiting secreted calpain or via an independent mechanism.

It is also intriguing why the OC-inhibitory effects of extracellular calpastatin, osteocyte-derived or recombinant, are more pronounced in females. Further, the in vivo phenotype of fewer OCs and enhanced bone mass was also only observed in female *Camkk2^ΔOCY^* mice. It is likely that sex-divergent signaling mechanisms downstream of secreted calpastatin in female and male OC precursors are responsible for this dichotomy. On the other hand, our proteomics analyses also identified 116 other proteins to be uniquely altered in the female *Camkk2^ΔOCY^* osteocyte secretome. Some of these proteins may also play OC-inhibitory roles, potentially collaborating with secreted calpastatin.

In conclusion, we identified a novel cell-intrinsic role for CaMKK2 in osteocytes that results in paracrine effects on OC formation and activity leading to enhanced bone mass only in female mice. This phenotype is in part due to a novel mechanism wherein extracellular calpastatin secreted by osteocytes inhibits bone resorption by female OCs. This novel osteocyte-secreted-calpastatin mechanism could be therapeutically leveraged to treat osteoporosis in women.

## 4. Materials and Methods

### 4.1. Mice 

All animal procedures were performed with prior approval from Indiana University School of Medicine Institutional Animal Care and Use Committee (IACUC). All experiments were performed in compliance with NIH guidelines on the use and care of laboratory and experimental animals. All animals were housed in the Indiana University School of Medicine Laboratory Animal Resource Center (LARC, Indianapolis, IN, USA) under a 12-h light, 12-h dark cycle. Food and water were provided ad libitum. All mice generated in this study were derived of C57BL/6J background. *Dentin matrix protein* (*Dmp1)-8kb*-*Cre^+^* mice [18,19] were provided by Dr. Teresita Bellido and *Camkk2^flox/flox^* mice have been described previously [11]. We first generated *Dmp1-Cre-Camkk2^flox+/^*^-^ by breeding the *Dmp1-Cre* and *Camkk2^fl/fl^* mice. Then we crossed the heterozygous mice to generate *Dmp1-Cre::Camkk2^+/+^* (*Control*) mice and *Dmp1-Cre::Camkk2^fl/fl^ (Camkk2^OCY^)* mice (Figure 1A). *Dmp1-Cre^+^::Camkk2^+/+^* mice were used to control for potential non-specific effects of *Dmp1*- regulated Cre recombinase. For all skeletal phenotyping and osteocyte isolation experiments, mice of either sex were used at 12 weeks (w) of age. Bone marrow isolations were performed using 6-week-old wild-type (WT) mice.

### 4.2. µCT Imaging 

Long bones were excised at 12 weeks of age and fixed in 4% paraformaldehyde (PFA) for 48 h at 4 °C and transferred to 70% ethanol. Micro-computed tomography (μCT) was performed on femurs at a 5.87 µm image pixel size using a Bruker 1172 μCT system (59 kV, 167 µA, 0.7 rotation step, 0.5 aluminum filter). Reconstructed μCT images (NRecon software, Kontich, Belgium) were analyzed using CT Analyzer software (Skyscan, Kontich, Belgium). The trabecular bone compartment was analyzed within 1 mm proximal of the distal growth plate, while femoral midshaft architecture was measured using cross-sections approximately 3.5 mm proximal to the distal growth plate. Reconstructed 3D models were generated using CTVox software (Skyscan, Kontich, Belgium), from which trabecular bone volume per total volume (BV/TV) (%),trabecular number (Tb.N) (mm^−1^), trabecular thickness (Tb.Th) (mm), and trabecular separation (Tb.Sp) (mm) as well as cortical bone parameters were calculated using established guidelines [49].

### 4.3. Histology 

Bone histology was performed by the Indiana Center for Musculoskeletal Health Musculoskeletal Histology Core. Following μCT analysis, dynamic and static histomorphometry were performed on undecalcified femurs embedded in poly-methyl methacrylate (plastic). Longitudinal sections cut at 5 µm thickness were stained with Von Kossa and McNeal’s (VKM) to assess OB numbers and osteoid parameters. Sections were also stained for tartrate-resistant acid phosphatase (TRAP) activity following with hematoxylin counterstain to measure OC parameters. Bone formation and mineralization parameters were measured based on incorporation of dual fluorochrome labels of calcein and alizarin, which mice received via I.P. injection approximately 7 and 2 days prior to euthanasia. Parameters measured included single-label perimeter (sL.Pm), double-label perimeter (dL.Pm), and interlabel width (Ir.L.Wi). From these primary measurements, the following outcome parameters were calculated: mineral apposition rate (MAR = Ir.L.Wi/7 days [µm/day]); mineralizing surface (MS/BS = (0.5 ∗ sL.Pm + dl.Pm)/B.Pm ∗100 [%]); and bone formation rate (BFR/BS = MAR ∗ MS/BS ∗ 365 [µm^3^/µm^2^/year]). Static and dynamic measurements and calculations followed the guidelines of the American Society for Bone and Mineral Research Histomorphometry Nomenclature Committee [50].

### 4.4. Immunohistochemistry 

Long bones were decalcified in 14% EDTA (pH 7.4) and dehydrated before paraffin embedding. Five µm thick serial sections were generated, deparaffinized and incubated with primary antibody against CaMKK2 (rabbit anti-CaMKK2-NT; cat# 033168, US Biological, Salem, MA, USA; 1:100 dilution) and anti-rabbit secondary (Jackson Immunoresearch, West Grove, PA, USA) and counterstained with Gill No. 1 Hematoxylin or Methyl Green (Millipore Sigma, Burlington, MA, USA). Images captured using a Leica DMi8 microscope were processed with Leica LAS-X software (Leica CM1950, Wetzlar, Germany), and quantified by counting immunopositive and total osteocytes within cortical bone from 4 regions of interest per sample using ImageJ (NIH, Bethesda, MD, USA).

### 4.5. Osteocyte Culture and Media Collection 

Osteocyte-enriched fractions derived from mouse long bones were isolated as previously reported by Stern et al. [51]. Cell suspensions from individual mice were cultured in twelve-well plates coated with type-I rat tail collagen (Millipore Sigma) at a seeding density of 1 × 10^5^ cells per well. In α-minimal essential medium supplemented with 2.5% fetal bovine serum (FBS), 2.5% bovine calf serum (BCS) and 1% penicillin and streptomycin (PS) (Thermo Fisher, Hampton, NH, USA). Cells were maintained at 37 °C and 5% CO_2_ in a humidified incubator. Osteocytes were not disturbed for the first 7 days while they attached to the plate. Media changes were performed every 48 h beginning on day 7, and conditioned media was collected from osteocyte cultures derived from individual mice every 48 h on days 9, 11, 13, and 15 post isolation. Conditioned media (CM) was filtered using a 0.22 µm syringe filter and aliquoted before storing at −20 °C. Serum-free CM was collected for mass spectrometry and immunodepletion assays after a 4 h incubation on day 7 of cultures.

### 4.6. Osteoclast Cultures and Media Supplementation: 

Bone marrow cells (BMCs) were isolated from the long bones of 6-week-old WT mice and plated on 0.1% gelatin-coated dishes at a density of 1.5 × 10^5^ cells/cm^2^ in osteoclast differentiation media (α-Minimum Essential Media (Invitrogen, Thermo Fisher) containing 10% FBS (R&D Systems, Minneapolis, MN, USA), P/S (Invitrogen), 30 ng/mL M-CSF (R&D Systems), and 50 ng/mL RANKL (Peprotech, Rockhill, NJ, USA), which was supplemented with 50% osteocyte conditioned media. BMCs were plated on gelatin-coated wells or glass coverslips to observe OC differentiation and actin-ring formation, respectively, whereas 96-well Corning^®^ Osteo Assay Surface strips (hydroxyapatite-coated, Corning, Thermo Fisher) were used to assess osteoclast resorption. Recombinant human calpastatin domain I (non-cell permeable) was purchased from Miilipore Sigma (catalogue #: 208900), reconstituted in sterile PBS and diluted before adding to OC cultures at 0.1 µM, 0.5 µM, 1.0 µM, and 5.0 µM final concentrations. OC precursor migration was observed by performing a controlled scratch assay after 3 days of differentiation. A sterile 10 µL pipette tip was used to scrape across the middle of each well and cell migration was observed over the course of 24 h. TRAP activity was assessed after 7 days of differentiation using the Acid Phosphatase Kit (Sigma). Actin ring formation was observed after 7 days by staining OCs mounted on glass coverslips with phalloidin-rhodamine (1:50; Thermo Fisher) diluted in PBS containing 1% BSA and mounted on slides using ProLong™ Gold Antifade mounting solution with DAPI (Invitrogen). To measure OC-mediated resorption after 8 days of differentiation, cells were removed from the 96-well Osteo Assay strip surface by incubating with 10% bleach solution for 5 minutes at room temperature, and resorption areas were visualized by Von Kossa staining. Resorbed areas appeared clear whereas the remaining hydroxyapatite-coated areas stained black with Von Kossa stain. The number of multinuclear (>3 nuclei) OCs as well as the number of resorption pits and resorption area were measured using Image J (NIH, USA).

### 4.7. qRT-PCR 

Primary osteocyte RNA was isolated using the RNAqueous kit (Invitrogen, Thermo Fisher). Samples were treated with DNAse I for 20 min at 37 °C to remove genomic DNA contamination and DNAse I was inactivated before synthesizing cDNA. RNA concentrations were determined using the BioPhotometer spectrophotometer (Eppendorf, Hauppauge, NY, USA). cDNA was synthesized from 1 μg RNA using a high-capacity reverse-transcriptase kit (Invitrogen, Thermo Fisher). QPCR reactions were performed using iTaq Universal SYBR Green Supermix and the CFX Connect™ Real-Time PCR Detection System (Bio-Rad Laboratories, Hercules, CA, USA). Relative expression to β-*Actin* was determined via the 2^−ΔΔCt^ method included in the CFX Manager Software (Bio-Rad Laboratories). Primers used in qPCR reactions were purchased from IDT (Integrated DNA Technologies, Coralville, IA) and are as follows: *Actin*-F (5′GGGAAATCGTGCGTGACATC), *Actin*-R (5′CCAAGAAGGAAGGCTGGAAAAG), *Rankl*-F (5′ CATTTGCACACCTCACCATCAAT), *Rankl*-R (5′GTCTGTAGGTACGCTTCCCG), *Opg*-F (5′ TCCCGAGGACCACAATGAACAAGT), *Opg*-R (5′TTAGGTAGGTGCCAGGAGCACATT), *Camkk2*-F (5′CATGAATGGACGCTGC) and *Camkk2*-R (5′TGACAACGCCATAGGAGCC).

### 4.8. Mass Spectrometry 

Serum-free osteocyte CM was concentrated 13-fold using centrifugal filter columns (Amicon^®^ Ultra 3K; Millipore Sigma), and protein concentrate was exchanged into ammonium bicarbonate buffer (25 mM). Samples were then dried down and resuspended in 8 M urea with 100 mM ammonium bicarbonate, pH 7.8. Disulfide bonds were reduced by incubation for 45 min at 57 °C with a final concentration of 10 mM Tris (2-carboxyethyl) phosphine hydrochloride (Catalog no C4706, Sigma Aldrich, St. Louis, MO, USA). A final concentration of 20 mM iodoacetamide (Catalog no I6125, Sigma Aldrich) was then added to alkylate these side chains and the reaction was allowed to proceed for one hour in the dark at 21 °C. Samples were diluted to 1 M urea using 100 mM ammonium bicarbonate, pH 7.8. One µg of trypsin (V5113, Promega, Madison, WI, USA) was added, and the samples were digested for 14 h at 37 °C. The following day, samples were desalted using Omix tips (Agilent, Santa Clara, CA, USA).

Peptide samples were analyzed by LC-MS on an Orbitrap Fusion Lumos (Thermo Fisher) equipped with an Easy NanoLC1200 HPLC (Thermo Fisher). Peptides were separated on a 75 µm × 15 cm Acclaim PepMap100 separating column (Thermo Scientific) downstream of a 2 cm guard column (Thermo Scientific). Buffer A was 0.1% formic acid in water. Buffer B was 0.1% formic acid in 80% acetonitrile. Peptides were separated on a 120 min gradient with the primary separation from 4% B to 33% B. Precursor ions were measured in the Orbitrap with a resolution of 120,000. Fragment ions were measured in the Orbitrap with a resolution of 15,000. The spray voltage was set at 1.8 kV. Orbitrap MS1 spectra (AGC 4 × 10^5^) were acquired from 400–2000 m/z followed by data-dependent HCD MS/MS (collision energy 30%, isolation window of 2 Da) for a three-second cycle time. Charge state screening was enabled to reject unassigned and singly charged ions. A dynamic exclusion time of 60 s was used to discriminate against previously selected ions.

### 4.9. Proteomics Data Analysis 

Resulting RAW files were analyzed in Proteome Discover™ 2.4.1.15 (Thermo Fisher Scientific, RRID: SCR_014477) with a *Mus musculus* UniProt FASTA (both reviewed and unreviewed sequences) plus FBS and common laboratory contaminants. Default minora feature detector settings were used. SEQUEST HT searches were conducted with a maximum number of 2 missed cleavages; precursor mass tolerance of 10 ppm; and a fragment mass tolerance of 0.02 Da. Carbamidomethylation on cysteine (C) residues was included as a static modification. Dynamic modifications used for the search were oxidation of methionines, phosphorylation on S, T, and Y, Met-loss or Met-loss plus acetylation of protein of N-termini. Percolator False Discovery Rate was set to a strict setting of 0.01 and a relaxed setting of 0.05. Quantification methods utilized feature mapper chromatographic alignment tools of maximum shift 10 and min s/n threshold of 5. Precursor ion quantification used precursor intensities of unique and razor peptides without scaling and with normalization based on total peptide amount. Quantitative rollup of the summed abundances was done with the top 3 N unmodified peptides, no imputation with a hypothesis of background-based *t*-tests. Resulting normalized abundance values for each sample type, abundance ratio and log2 (abundance ratio) values; and respective *p*-values from Proteome Discover™ were exported to Microsoft Excel, Uniprot accession numbers were uploaded to the online Database for Annotation, Visualization and Integrated Discovery (DAVID) to perform a functional annotation analysis. All processed and raw data are available upon request.

### 4.10. Calpastatin Enzyme-Linked Immunosorbent Assay (ELISA) 

Quantification of calpastatin levels in osteocyte CM was performed using the Mouse CAST (Calpastatin) ELISA kit (*NBP2-75048*; Novus Biologicals, Centennial, CO, USA) according to manufacturer’s directions.

### 4.11. Immunoblotting 

Primary osteocytes or WT OCs were harvested and placed into iced tween lysis buffer (25 mM Hepes (pH 7.5), 50 mM NaCl, 25 mM NaH_2_PO_4_, 0.5% Tween 20, 10% glycerol, 1 mM dTT, 10 mM β-glycerophosphate, 2 mM EGTA, 2 mM EDTA, 25 mM NaF, 1 mM sodium vanadate, 1 mM PMSF, 1 mg/mL aprotinin, 1 mg/mL leupeptin, 10 mg/mL pefabloc, and 100 nM okadaic acid), sonicated on ice, centrifuged 14,000 rpm for 30 min at 4 °C. Equal amount of protein lysates (7.5 µg/lane) isolated from primary osteocytes or osteoclasts was fractionated under denaturing conditions on SDS-PAGE and transferred onto Immobilon-P membranes (Millipore Sigma). Blocking, primary and secondary antibody incubations were performed in Tris-buffered saline (TBS) containing 5% non-fat dry milk. Washes were performed in TBS with Tween-20 (0.1%, *v*/*v*, Millipore Sigma). Membranes were probed with primary antibodies for Cast (Cell Signaling Technology, Danvers, MA, USA; 1:1000), CaMKK2 (BD Biosciences; 1:1000), Talin (C-9) (Santa Cruz Biotechnology, Dallas, TX, USA; SC-365875 1:100), Rabbit p-PKA-C (T197), total PKA-C (D45D3) (both Cell Signaling 1:1000) or b-Actin (MilliporeSigma; 1:3000) followed by horseradish peroxidase-conjugated secondary antibodies (Jackson Immunoresearch; 1:5000). The target proteins were visualized with Clarity Enhanced Chemiluminescence substrate (Bio-Rad) using a ChemiDox MP Image System (Bio-Rad), and band densities quantified using ImageJ.

### 4.12. Immunoprecipitation of Calpastatin from Conditioned Media 

Osteocyte CM used directly for Western blot or media supplementation for osteoclast assays was precleared with Protein G Magnetic Sepharose Xtra Beads (GE Healthcare) and incubated with either rabbit anti-Cast antibody (1:1000, Cell Signaling #4416) or control IgG (1:1000, MilliporeSigma) overnight on a slow rotating mixer at 4 °C. Protein G Magnetic Sepharose Xtra beads were added and slurry allowed to incubate for 1 h with rotation at 4 °C, and centrifuged to separate immunoprecipitate containing beads from supernatant. For media supplementation, the supernatant was passed through a 0.22 µm syringe filter and aliquoted for storage at −20 °C. For immunoblots, beads containing the immunoprecipitates were washed three times with DPBS at room temperature and boiled in 1× protein loading dye prior to loading (30 µg/lane) on SDS-PAGE gel (Novex) for immunoblotting with Rabbit α-Calpastatin antibody (Cell signaling 1:1000).

### 4.13. Statistical Analysis 

Statistical analyses were performed using the GraphPad Prism software (GraphPad Software, San Diego, CA, USA). Normality assumptions were evaluated using histograms and QQ plots. Data sets that passed normality test were analyzed using unpaired, two-tailed Student’s *t*-test when comparing Control and *Camkk2^OCY^* or ordinary one-way ANOVA followed by Tukey’s post-hoc analysis when comparing >2 groups [52]. When data sets did not pass the normality test, non-parametric tests were used: Mann–Whitney test for 2 sample comparisons or Kruskal–Wallis test followed by Dunn’s post hoc to compare >2 groups). All values are represented as mean ± standard deviation (SD).

## Figures and Tables

**Figure 1 ijms-24-04718-f001:**
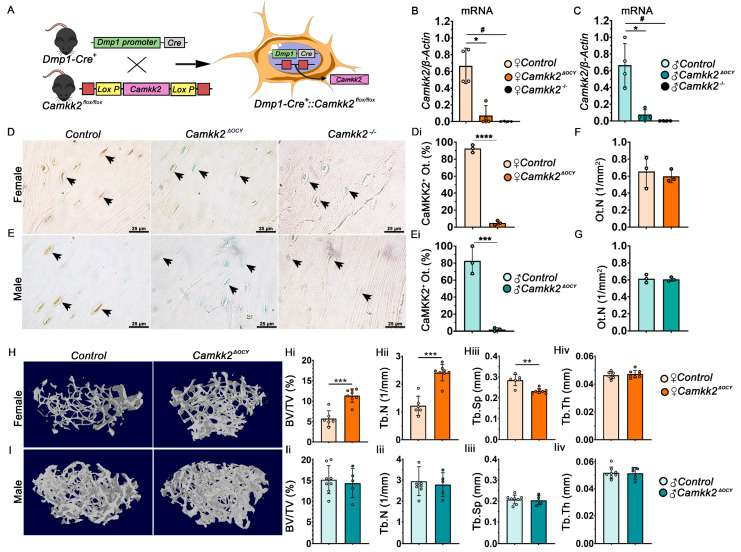
CaMKK2 deficiency in osteocytes enhances trabecular bone mass in female but not male mice. (**A**) Diagram illustrating the breeding scheme for the generation of the mouse model with osteocyte-targeted deletion of CaMKK2. (**B**,**C**) Normalized *Camkk2* expression relative to *β-actin* in primary osteocytes isolated from the long bones of female and male control, *Camkk2^ΔOCY^* and *Camkk2^-/-^* mice (*n* = 4 individual mice/group). (**D**,**E**) Representative IHC images (*n* = 3; magnification X 630) of CaMKK2 immunostaining in femoral cortical bone of female and male control *(Camkk2^WT/WT^)*, *Camkk2^ΔOCY^* and *Camkk2^-/-^* mice. (**Di**,**Ei**) CaMKK2-immunopositive osteocytes (CaMKK2^+^ Ot) shown as percentage of total osteocytes. (**F**,**G**) Total number of osteocytes (N.Ot/mm^2^) enumerated from female and male control and *Camkk2^ΔOCY^* long bones (*n* = 3/group). (**H**,**I**) Representative 3D models of cancellous bone microarchitecture in the distal femur of female and male control and *Camkk2^ΔOCY^* mice. (**Hi**,**Ii**) Mean percent bone volume (BV/TV); (**Hii**–**Hiv**,**Iii**–**Iiv**) number of trabeculae (Tb.N), spacing between trabeculae (Tb.Sp), and thickness of trabeculae (Tb.Th) within the distal femur of female and male *control* and *Camkk2^ΔOCY^* mice (*n* = 5–8/group). Error bars represent SD. * denotes statistical comparisons between control and *Camkk2^ΔOCY^* mice; * *p* < 0.05, ** *p* < 0.01, *** *p* < 0.001, **** *p* < 0.0001. # = *p* < 0.05 – statistical comparisons between control *and Camkk2^-/-^* mice.

**Figure 2 ijms-24-04718-f002:**
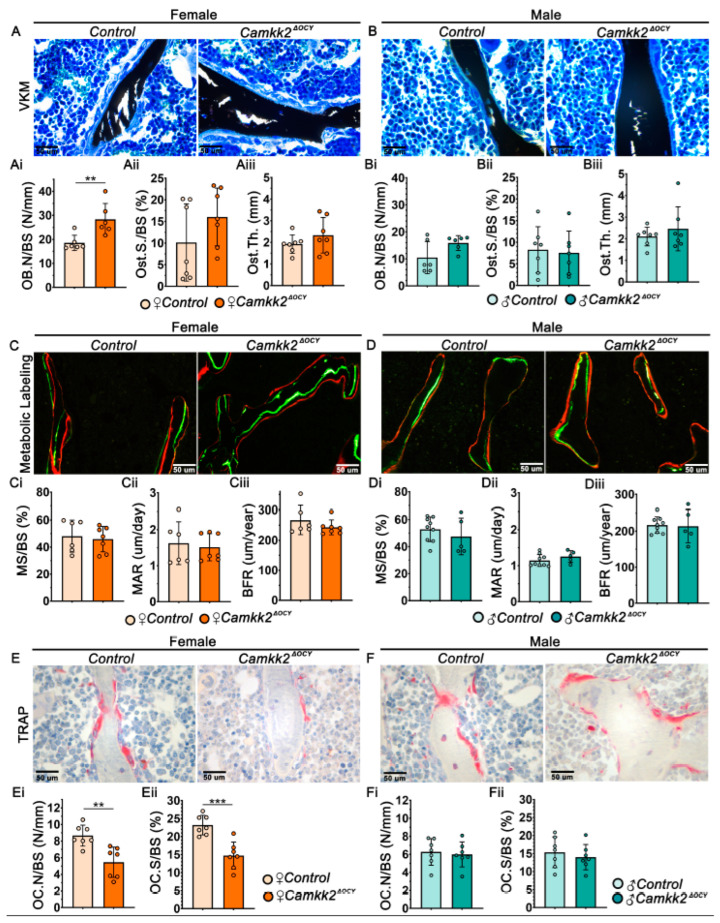
Altering osteocyte expression of CaMKK2 suppresses OCs without influencing OBs. (**A**,**B**) Representative images of female and male distal femur trabeculae stained with Von Kossa and McNeil’s tetrachrome (VKM; magnification 400×); mineralized bone = black; osteoid = light blue. (**Ai**,**Bi**) Mean osteoblast numbers (N.Ob/BS), (**Aii**,**Bii**) osteoid surface (OS/BS), and (**Aiii**,**Biii**) osteoid thickness (O.Th) measured among female and male control and *Camkk2^ΔOCY^* bone samples (*n* = 6–7/group). (**C**,**D**) Representative images (magnification 400×) of female and male trabecular bone mineralization following incorporation of calcein (green) and alizarin red fluorochrome labels to assess mineralized surface (MS/BS), mineral apposition rate (MAR), and bone formation rate (BFR) (**Ci–Ciii**,**Di–Diii**) respectively, in control and *Camkk2^ΔOCY^* long bones (*n* = 6–7/group). (**E**,**F**) Representative images (*n* = 7/group; magnification 400×) of female and male distal femur sections stained for TRAP activity in OCs in vivo. (**Ei**,**Fi**) Number of TRAP-positive OCs (red) attached to trabecular bone (N.Oc/BS) in femur sections. (**Eii**,**Fii**) Osteoclast surface per bone surface (OC.S/BS)—indicator of bone attachment by OCs. Error bars represent SD. ** *p* < 0.01, *** *p* < 0.001.

**Figure 3 ijms-24-04718-f003:**
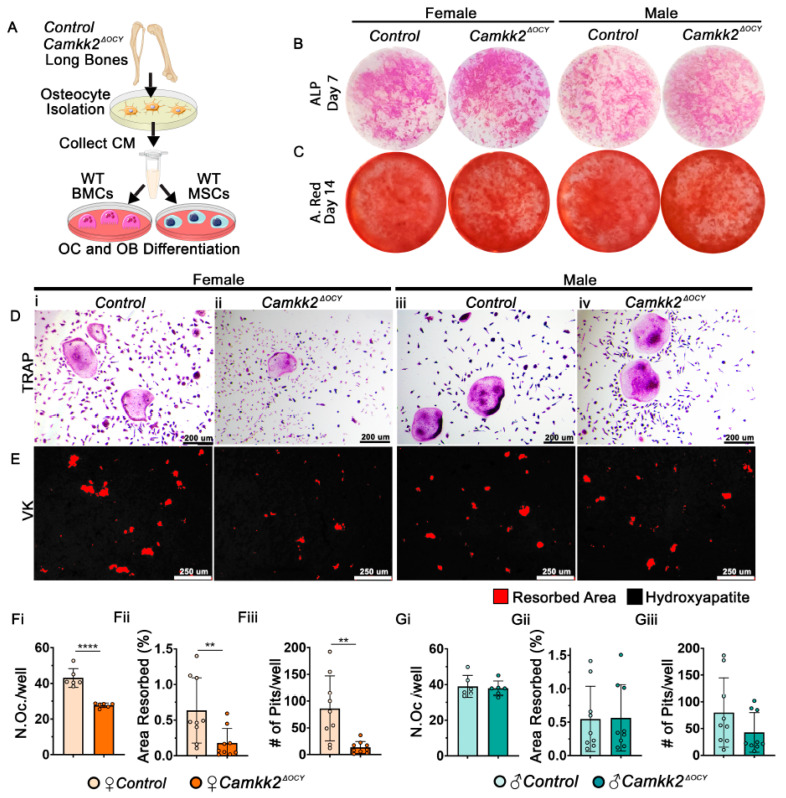
Conditioned media from female osteocytes lacking CaMKK2 inhibits OCs. (**A**) Schematic of in vitro experimental design for primary osteocyte isolation from long bones of individual mice and collection of osteocyte conditioned media (CM) to test OB differentiation by WT BM-derived MSCs and OC differentiation by WT BM cells (BMCs). CM from cells isolated from each mouse was kept separate and treated as a separate sample. (**B**,**C**) Representative images of alkaline phosphatase (**B**) and alizarin red (**C**) staining on days 7 and 14 of osteogenic differentiation, respectively (*n* = 3/group). (**D**) Representative images (*n* = 6/group; magnification 100×) of female and male TRAP-positive WT OCs (purple) treated with indicated osteocyte conditioned media. (**E**) Representative images (*n* = 9/group) of hydroxyapatite-coating (black) stained with Von Kossa (VK) to reveal areas of osteoclast-mediated resorption pits (pseudo-colored red). (**Fi**,**Gi**) Enumeration of TRAP-positive osteoclasts with ≥3 nuclei, (**Fii**,**Gii**) percent area resorbed and (**Fiii**,**Giii**) total number of pits measured using ImageJ. Error bars represent SD. ** *p* < 0.01, **** *p* < 0.0001.

**Figure 4 ijms-24-04718-f004:**
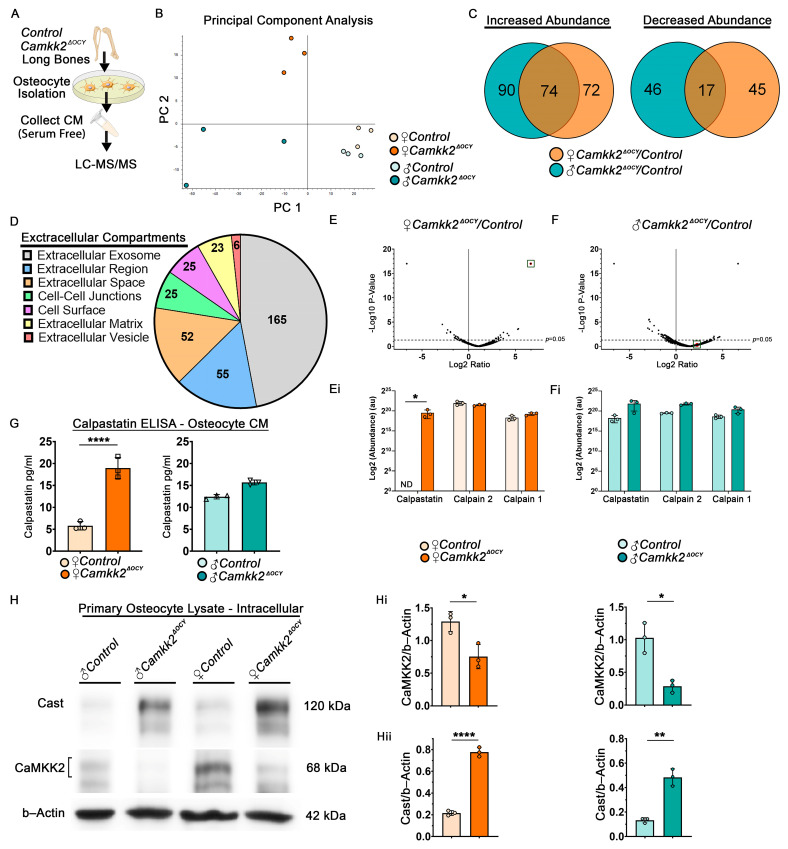
Calpastatin is enhanced in CaMKK2 deficient osteocyte conditioned media from female mice. (**A**) Scheme for primary osteocyte isolation from the long bones of individual female and male *control* and *Camkk2^ΔOCY^* mice and collection of serum-free conditioned media for LC-MS/MS analysis. (**B**) Principal component analysis (PCA) plot of proteomic signatures of osteocyte CM (*n* = 3/group). (**C**) Venn diagram of proteins with increased or decreased abundance in female and male *Camkk2^ΔOCY^* osteocyte CM relative to sex-matched controls. (**D**) Pie chart showing functional annotation of osteocyte CM proteins. All GO-Terms with a minimum False Discovery Rate (FDR) corrected *p*-value < 0.05 were included. (**E**,**F**) Volcano plots of enriched proteins in female and male *Camkk2^ΔOCY^* osteocyte CM relative to sex-matched controls. Calpastatin (Cast) is represented within the cluster colored as red (box). (**Ei**,**Fi**) Plots of Log2 (abundance) values for Cast, calpain (Capn)2 and Capn1 in female and male osteocyte CM. (**G**) Amount of calpastatin (pg/mL) in osteocyte CM in indicated CM samples measured using ELISA (*n* = 3/group). (**H**) Immunoblot of primary osteocyte lysates (*n* = 3/independent experiments) for the detection of intracellular Cast, CaMKK2 and b-Actin. (**Hi**,**Hii**) Quantification of intracellular CaMKK2 and Cast levels relative to b–Actin. Error bars represent SD. Abundance ratios above the line (*p* < 0.05) were deemed significant. * *p* < 0.05, ** *p* < 0.01, **** *p* < 0.0001.

**Figure 5 ijms-24-04718-f005:**
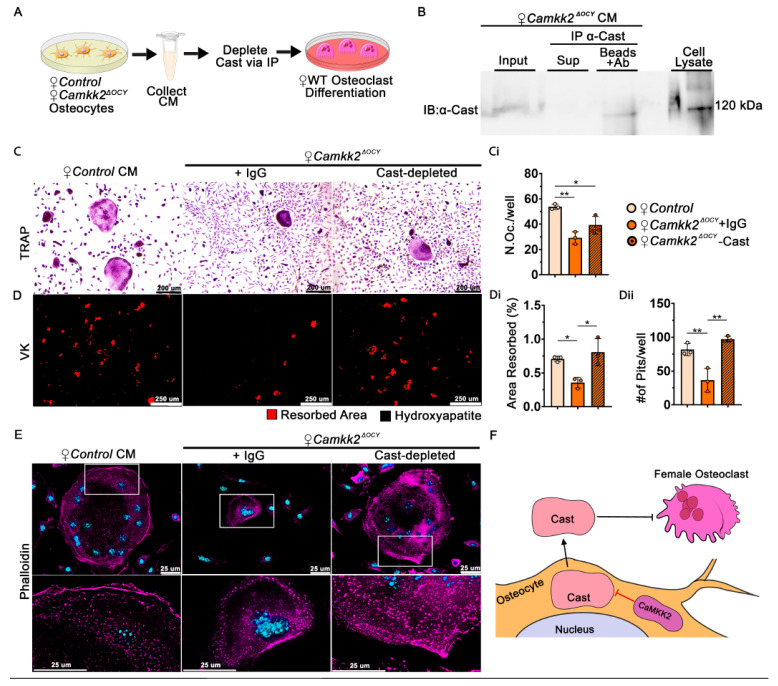
Depletion of Calpastatin from conditioned media of female CaMKK2-deficient osteocytes restores osteoclast function. (**A**) Schematic of experimental design outlining the collection of female osteocyte CM and depletion of Cast by immunoprecipitation (IP) with α-Cast antibody or control IgG prior to treating WT BM cells during OC differentiation. (**B**) Immunoblot of Cast levels in female *Camkk2^ΔOCY^* osteocyte CM (input) prior to IP as well as supernatant and substrate (Cast) captured by antibody conjugated beads post-IP. (**C**,**D**) Representative images (*n* = 3/group) magnification 100×) showing TRAP-positive OCs and hydroxyapatite resorption by WT OCs treated with female control osteocyte CM or *Camkk2^ΔOCY^* osteocyte CM depleted of Cast or treated with control IgG. (**Ci**) Number of TRAP positive OCs (purple) with 3 or more nuclei, (**Di**) percent hydroxyapatite area resorbed (red), and (**Dii**) number of individual pits measured using ImageJ (*n* = 3/group). (**E**) Representative images (magnification 630×) of phalloidin staining OCs from indicated treatment conditions to visualize the filamentous (**F**) actin cytoskeleton (pseudo-colored magenta) and DAPI stain to visualize nuclei (blue). Magnification (200%) of outlined regions of interest using Leica LASX software is shown underneath each image. (**F**) Proposed mechanism indicating that during homeostasis, CaMKK2 negatively regulates osteocyte Cast levels. Osteocyte-derived extracellular Cast inhibits the function of female OCs. Error bars represent SD. * *p* < 0.05, ** *p* < 0.01.

**Table 1 ijms-24-04718-t001:** Pathway Enrichment Analysis.

**Increased Protein Abundance (*Camkk2^ΔOCY^*/*Control*)**		**Female**		**Male**	
**Biological Process**	**Reactome Pathway**	**Fold Enrichment**	***p*-Value**	**Fold Enrichment**	***p*-Value**
Cell Cycle	Golgi Cisternae Pericentriolar Stack Reorganization	6.7	0.082	10.5	0.092
	CDK-mediated phosphorylation and removal of Cdc6	6.7	0.021	10.5	0.000046
	SCF(Skp2)-mediated degradation of p27/p21	N/A	N/A	10.3	0.00001
	Autodegradation of the E3 ubiquitin ligase COP1	6.6	0.022	10.3	0.000051
	Ubiquitin Mediated Degradation of Phosphorylated Cdc25A	6.5	0.023	10.1	0.000056
	AUF1 binds and destabilizes mRNA	6.3	0.025	9.8	0.000069
	Cdc20:Phospho-APC/C mediated degradation of Cyclin A	6.1	0.0087	N/A	N/A
Extracellular Matrix Organization	Degradation of the extracellular matrix	6	0.028	N/A	N/A
Hemostasis	Cell surface interactions at the vascular wall	7.4	0.016	N/A	N/A
Immune System	Cross-presentation of soluble endosomes	7.4	0.016	11.5	0.000026
	Activation of NF-kappaB	N/A	N/A	9.4	0.000019
Metabolism	Regulation of ornithine decarboxylase (ODC)	8.6	0.0026	10.7	0.000041
Signal Transduction	Degradation of GLI1 by the proteasome	6.1	0.027	10.8	0.0000074
	GLI3 is processed to GLI3R by the proteasome	N/A	N/A	10.6	0.0000082
	Dectin-1 mediated noncanonical NF-kB signaling	N/A	N/A	10.5	0.0000092
	NIK noncanonical NF-kB signaling	N/A	N/A	10.5	0.0000092
	Degradation of beta-catenin by the destruction complex	6.5	0.0069	10.4	0.0000018
	Degradation of AXIN	6.3	0.025	9.8	0.000069
	Degradation of DVL	6.1	0.027	9.4	0.000083
	Hedgehog ligand biogenesis	N/A	N/A	9.3	0.000091
Vesicle-Mediated Transport	Golgi Associated Vesicle Biogenesis	7.1	0.018	N/A	N/A
	COPI-mediated anterograde transport	6.5	0.0069	N/A	N/A
**Decreased Protein Abundance (*Camkk2^ΔOCY^*/*Control*)**		**Female**		**Male**	
**Biological Process**	**Reactome Pathway**	**Fold Enrichment**	***p*-Value**	**Fold Enrichment**	***p*-Value**
Metabolism of Proteins	Formation of a pool of free 40S subunits	8.1	0.0012	N/A	N/A
	Formation of the ternary and 43S complex	6.5	0.0015	N/A	N/A
	L13a-mediated translational silencing of Ceruloplasmin expression	8.1	0.0017	N/A	N/A
	GTP hydrolysis and joining of the 60S ribosomal subunit	6.5	0.0021	N/A	N/A
	Translation initiation complex formation	6.5	0.0022	N/A	N/A
	Ribosomal scanning and start codon recognition	6.5	0.009	N/A	N/A
	SRP-dependent co-translational protein targeting to membrane	6.5	0.0093	N/A	N/A
Metabolism of RNA	Nonsense Mediated Decay enhanced by the Exon Junction Complex	6.5	0.016	N/A	N/A
Signal Transduction	Signaling by EGFR	N/A	N/A	77.3	0.025
	EGFR Transactivation by Gastrin	N/A	N/A	60.1	0.032
Cell-Cell Communication	Type I hemidesmosome assembly	N/A	N/A	49.2	0.039
	Adherens junctions interactions	N/A	N/A	26.2	0.0053
Extracellular matrix Organization	Collagen degradation	N/A	N/A	15.9	0.0017
	Degradation of the extracellular matrix	N/A	N/A	13.1	0.02

## Data Availability

The data presented in this study are available on request from the corresponding author. Raw data from our proteomics study has been uploaded onto Mendeley data base—https://data.mendeley.com/datasets/pnt97mzggk/1 (access on 31 March 2020) and are available on request from the corresponding author.

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
