# Peer review of "Osteocyte-Derived CaMKK2 Regulates Osteoclasts and Bone Mass in a Sex-Dependent Manner through Secreted Calpastatin"

_ijms, 2023, doi:10.3390/ijms24054718_

Round 1

Reviewer 1 Report

Osteocyte-Derived CaMKK2 Regulates Osteoclasts and Bone Mass in a Sex-Dependent Manner Through Secreted Calpastatin

In this manuscript, the authors tested the hypothesis that osteocyte derived CaMKK2 plays an important

role in the regulation of bone growth and maintenance using in vivo and in vitro approaches. Their data revealed a novel paracrine role for osteocyte-derived CaMKK2 in the regulation of OC differentiation and function through a secreted calpastatin-mediated mechanism.

The introduction of the manuscript is relevant and gives sufficient information about the previous studies findings which leads to the current study rationale. To accomplish the aim of this study, the authors used many in vivo and in vitro techniques. The methodology is generally appropriate, well-presented and organized in a logical way, beginning with the choice of a model, and continuing on through the selection of procedures, stains, analytical tools, and statistical methods. They concluded their work by recommending the novel osteocyte-secreted-calpastatin mechanism as a therapeutically leveraged to treat osteoporosis in women.

The study is so interesting, well-designed, and the manuscript is clear and well-written. However, the authors need to give more clarifications about the following comments:

Major points:

There are no figures attached to the manuscript (only figure 4H, 5B, and supplementary are attached).

Minor points: 

In the results section, you mentioned that 62 proteins are highly enriched in CM from Camkk2ΔOCY osteocytes. Is this the only protein that can be selected for further study?, if not why you only assessed this protein.

For supplementary figure 3B-G, please put titles for B-G and E-G with the name of the groups to make the figure understandable by itself.

Author Response

The study is so interesting, well-designed, and the manuscript is clear and well-written.

  • We thank the reviewer for their enthusiasm for our study. We have tried to address all their concerns to our best ability by revising the supplementary figure or adding an explanation as seen below. Hope the reviewer is satisfied with our actions.

However, the authors need to give more clarifications about the following comments:

Major points:

There are no figures attached to the manuscript (only figure 4H, 5B, and supplementary are attached).

  • We apologize for this confusion. We hope that the issue was resolved in time for the reviewer.   

Minor points: 

In the results section, you mentioned that 62 proteins are highly enriched in CM from Camkk2ΔOCY osteocytes. Is this the only protein that can be selected for further study? if not why you only assessed this protein?

  • Calpastatin was among a group of 62 proteins that were highly enriched in conditioned media from female Camkk2ΔOCY osteocytes, but not detected in that from female control osteocytes. Of the proteins associated with extracellular matrix degradation, calpastatin was of particular interest to us. This is because, like CaMKK2, calpastatin activity is regulated by Ca2+ and it possesses calmodulin (CaM) binding domains. Moreover, the prime target calpain has crucial roles in osteoclasts, and calpain is a Ca2+-dependent cysteine protease. We were especially intrigued because although predominantly intracellular, calpastatin and calpains are also secreted molecules, and the function of extracellular calpastatin in the bone is unknown.  We have attempted to clarify these points in pages 14-15 of the manuscript.  We hope the reviewer is satisfied with our explanation.  

For supplementary figure 3B-G, please put titles for B-G and E-G with the name of the groups to make the figure understandable by itself.

  • We revised supplementary figure 3 per the reviewer’s suggestion.

Reviewer 2 Report

In this study, Williams et al. investigated osteocyte-specific role of CaMKK2 by generating osteocyte-specific CaMKK2 deletion mice (Camkk2DOCY). The authors found that CaMKK2-deficiency in osteocytes resulted in sex-specific (female specific) increase in bone mass, and decreased osteoclast number and bone resorbing activity. The authors found that conditioned medium of in vitro cultured osteocytes from female Camkk2DOCY mice inhibited wild-type osteoclast differentiation but not osteoblast differentiation, suggesting that osteocytes from CaMKK2-deficient female mice secret factor(s) which could inhibit osteoclasts. To gain insight into this phenotype, the authors collected conditioned medium from male and female control and Camkk2DOCY mice, and performed proteomic analysis. This analysis identified 117 proteins significantly altered only in osteocytes from female Camkk2DOCY mice. The authors focused on calpastatin, a protein associated with ECM degradation, regulated by Ca2+, and known to have crucial roles in osteoclasts. The authors confirmed higher levels of calpastatin in female Camkk2DOCY conditioned medium than control conditioned medium. In vitro RANKL-induced osteoclast formation was inhibited by recombinant human calpastatin in a dose-dependent manner, and the inhibitory effect was more pronounced on female BMs than male BMs. Depletion of calpastatin in conditioned medium by using antibody against calpastatin abolished inhibitory effect of conditioned medium on osteoclast formation. Finally, the authors investigated the inhibitory effect of calpastatin on osteoclast formation, and showed that calpastatin did not inhibit cell migration but inhibited actin ring formation.

This is a very interesting study. Experiments were carefully conducted, results were convincing, and text was carefully and well written. Some questions were left, but the authors sincerely discussed about them. I believe that this study will contribute to a lot of researchers in this research field. I have just a few questions/comments.

In figure 2, female Camkk2DOCY mice have increased osteoblast numbers but have comparable bone formation rate with control mice. Does this mean that bone forming function of osteoblast per se in Camkk2DOCY female mice was weaker than control osteoblasts, but compensate the weaker bone formation by increasing osteoblast numbers?

In figure 5, cast-depletion resulted in partial recovery (it seemed not full but partial, to me) of osteoclast formation, but resulted in full recovery or even enhanced osteoclast resorbing function. How do I interpret this result?

Table I, it would be better to show fold-enrichment value of each Reactome pathway.

In page 15, line 6-9 from the top: this sentence lacked the word “male”.

Author Response

This is a very interesting study. Experiments were carefully conducted, results were convincing, and text was carefully and well written. Some questions were left, but the authors sincerely discussed about them. I believe that this study will contribute to a lot of researchers in this research field.

  • We thank the reviewer for their enthusiasm for our study and their positive comments. We have tried to address all their concerns to our best ability by revising the Table or adding an explanation as seen below. Hope the reviewer is satisfied with our actions.

I have just a few questions/comments.

In figure 2, female Camkk2DOCY mice have increased osteoblast numbers but have comparable bone formation rate with control mice. Does this mean that bone forming function of osteoblast per se in Camkk2DOCY female mice was weaker than control osteoblasts, but compensate the weaker bone formation by increasing osteoblast numbers?

  • We understand the reviewer’s concern and agree that it is a possibility that osteoblast function could be altered in the 12-week-old female Camkk2ΔOCY mice used in this study. However, we have measured the same metrics using older (24-week-old) female mutant mice for another project and did not observe any such differences. Since at 12 weeks, mice are still growing, albeit at a lower rate, we concluded that the differential osteoblast numbers observed in these mice could be an age-associated factor.  Nevertheless, we are confident of our results as the data were collected in a blinded manner, independently by two individuals and from n=6-7 mice per group.

In figure 5, cast-depletion resulted in partial recovery (it seemed not full but partial, to me) of osteoclast formation, but resulted in full recovery or even enhanced osteoclast resorbing function. How do I interpret this result?

  • We apologize for not being clearer on this. We agree that depletion of calpastatin partially rescues osteoclast formation and fully rescues resorption.  We interpret this to mean that secreted calpastatin is a potent inhibitor of female osteoclast resorbing function.  Data presented in Supplementary Figure 3 also support this interpretation as recombinant non-cell permeable (NCP) calpastatin suppresses female osteoclast function more significantly in a dose-dependent manner than male osteoclasts, the function of which is only suppressed at highest concentration of 5 µM recombinant NCP calpastatin.  Furthermore,  we observed a 2.3-fold suppression of female osteoclast resorption pit number and area, but only a 1.2-fold inhibition of osteoclast formation by 0.5 µM recombinant NCP calpastatin.  At this time, we do not fully understand the underlying mechanism of female osteoclast resorption function inhibition by extracellular calpastatin but are actively investigating this through another project.  We hope the reviewer is satisfied with our explanation.

Table I, it would be better to show fold-enrichment value of each Reactome pathway.

  • We agree with the reviewer that there is value in adding the fold-enrichment information to the table showing pathway enrichment analysis, and the new table 1 in the revision (pages 13-14) includes fold-enrichment values.

In page 15, line 6-9 from the top: this sentence lacked the word “male”.

Done

Round 2

Reviewer 1 Report

I would like to thank the authors as they addressed all the suggested comments.